# Anti-CD26 Antibody Suppresses Epithelial-Mesenchymal Transition in Colorectal Cancer Stem Cells

**DOI:** 10.3390/ijms26157620

**Published:** 2025-08-06

**Authors:** Takumi Iwasawa, Ryo Hatano, Satoshi Takeda, Ayumi Kurusu, Chikako Okamoto, Kazunori Kato, Chikao Morimoto, Noriaki Iwao

**Affiliations:** 1Shizuoka Medical Research Center for Disaster, Juntendo University, 1129 Nagaoka Izunokuni, Shizuoka 410-2295, Japan; iwasawa@toyo.jp (T.I.);; 2Institute of Life Innovation Studies, Toyo University, Tokyo 115-8650, Japan; 3Department of Therapy Development and Innovation for Immune Disorders and Cancers, Juntendo University, 2-1-1 Hongo Bunkyo Ward, Tokyo 113-8421, Japan; 4Atopy Research Center, Graduate School of Medicine, Juntendo University, 2-1-1 Hongo Bunkyo Ward, Tokyo 113-8421, Japan; 5Division of Blood Transfusion, Juntendo Shizuoka Hospital, 1129 Nagaoka Izunokuni, Shizuoka 410-2295, Japan

**Keywords:** CD26, EMT, colorectal cancer, cancer stem cell, antibody drug

## Abstract

CD26 (dipeptidyl peptidase-4) is a marker of colorectal cancer stem cells with high metastatic potential and resistance to therapy. Although CD26 expression is known to be associated with tumor progression, its functional involvement in epithelial-mesenchymal transition (EMT) and metastasis remains to be fully elucidated. In this study, we aimed to investigate the effects of a monoclonal anti-CD26 antibody on EMT-related phenotypes and metastatic behavior in colorectal cancer cells. We evaluated changes in EMT markers by quantitative PCR and Western blotting, assessed cell motility and invasion using scratch wound-healing and Transwell assays, and examined metastatic potential in vivo using a splenic injection mouse model. Treatment with the anti-CD26 antibody significantly increased the expression of the epithelial marker E-cadherin and reduced levels of EMT-inducing transcription factors, including ZEB1, Twist1, and Snail1, at the mRNA and protein levels. Functional assays revealed that the antibody markedly inhibited cell migration and invasion in vitro without exerting cytotoxic effects. Furthermore, systemic administration of the anti-CD26 antibody significantly suppressed the formation of liver metastases in vivo. These findings suggest that CD26 may contribute to the regulation of EMT and metastatic behavior in colorectal cancer. Our data highlight the potential therapeutic utility of CD26-targeted antibody therapy for suppressing EMT-associated phenotypes and metastatic progression.

## 1. Introduction

Colorectal cancer (CRC) remains a leading cause of cancer-related mortality worldwide, with metastasis being the primary determinant of patient prognosis [1]. Increasing evidence suggests that the subpopulation of tumor cells known as cancer stem cells (CSCs) plays a pivotal role in tumor initiation, progression, resistance to therapy, and metastatic dissemination [2]. Among these, CD26 (dipeptidyl peptidase-4 [DPP-4]) has been identified as a functional marker for highly metastatic CSCs in CRC. Pang et al. reported that CD26^+^ cells derived from human CRC tissues exhibit enhanced migratory capacity and can initiate liver metastasis in vivo, implicating CD26 as a driver of metastatic potential in colorectal CSCs [3].

CD26 is a type II transmembrane glycoprotein with multifunctional roles, including enzymatic cleavage of dipeptides, modulation of chemokine activity, immune regulation, and cell adhesion [4]. Besides its enzymatic functions, CD26 acts as a signaling hub by interacting with various membrane proteins and extracellular matrix components, thereby affecting cell behavior in physiological and pathological contexts [5]. In CRC, CD26 expression correlates with tumor aggressiveness, resistance to chemotherapy, and poor clinical outcomes [6]. However, the precise molecular mechanisms through which CD26 contributes to metastatic progression remain unclear.

An important mechanism facilitating cancer metastasis is epithelial-mesenchymal transition (EMT), a dynamic process by which epithelial tumor cells acquire mesenchymal features, such as reduced cell–cell adhesion, increased motility, and invasiveness [7]. EMT is regulated by several transcription factors, including ZEB1, Snail1, and Twist1, and is frequently associated with the acquisition of stem-like properties and chemoresistance in cancer cells [8,9]. Notably, EMT is not only a driver of invasion and dissemination but also contributes to the maintenance of CSC phenotypes [10].

Given the central roles of CD26 and EMT in CRC metastasis, we hypothesized that CD26 may promote metastatic potential by regulating EMT-related signaling pathways. Therefore, in the present study, we aimed to investigate the effects of an anti-CD26 monoclonal antibody on EMT markers, metastasis-associated cellular behaviors, and in vivo metastatic capacity in a murine model.

## 2. Results

### 2.1. Structure of the CD26 Molecule and Cytotoxicity of Anti-CD26 Antibodies

To establish the molecular characteristics of the target antigen, we first illustrated the structural features of CD26 (DPP-4), a type II transmembrane glycoprotein composed of a short cytoplasmic tail, single transmembrane domain, and large extracellular region that includes the enzymatic active site and binding domains (Figure 1A). Figure 1A illustrates the antigen-binding regions and domains responsible for specific recognition of CD26, providing the basis for subsequent functional analyses. To further examine CD26 expression in clinical samples, we analyzed publicly available transcriptomic datasets using the GEPIA platform, which integrates TCGA and GTEx data. As shown in Figure 1B, CD26 expression was significantly higher in tumor tissues compared to normal tissues in both colon adenocarcinoma and rectum adenocarcinoma, supporting its relevance in colorectal cancer pathophysiology. To confirm the safety profile of the anti-CD26 antibody, we assessed its cytotoxicity using an AlamarBlue assay in HCT-116 CRC cells. Cells were treated with varying concentrations of the antibody, up to 75 μg/mL, for 48 h. The results demonstrated no significant reduction in cell viability at the highest tested concentration (75 μg/mL) compared with that of untreated control cells (Figure 1C). These data indicate that the anti-CD26 antibody does not exhibit substantial cytotoxic effects on CRC cells at therapeutically relevant concentrations, supporting its potential safety for further therapeutic investigations.

### 2.2. Expression Levels of CD26 and EMT-Related Molecules

To examine the relationship between CD26 expression and EMT markers, we compared the mRNA levels of CD26, ZEB1, and Twist1 in HCT-116 cells, cultured for four and seven days, using quantitative PCR. The results demonstrated that the expression levels of CD26, ZEB1, and Twist1 were significantly increased on day seven compared with those on day four (Figure 2A). Consistent with previous studies showing that CD26 expression increases following prolonged culture and confluence, our findings confirmed this trend [11]. In addition, because ZEB1 and Twist1 are established markers of EMT, the concurrent increase in their expression alongside CD26 suggests a potential association between elevated CD26 levels and the progression of EMT in CRC cells. Microscopic images taken on days four and seven showed that on day four, there were still a few gaps between the cells, but on day seven, the cells were spread across the entire field of view (Figure 2B).

### 2.3. Anti-CD26 Antibody Treatment Inhibits Migration and Invasion of HCT-116 Cells

To evaluate the effect of the anti-CD26 antibody on cell migration, we performed a scratch wound-healing assay using HCT-116 CRC cells. Treatment with the anti-CD26 antibody significantly suppressed the migration of HCT-116 cells into the wound area compared with that of the control group (Figure 3A,B). Furthermore, the migration of HCT-116 cells was significantly inhibited according to Transwell assay results (Figure 3C). This reduction in cell motility is likely attributable to the inhibition of EMT, consistent with the observed downregulation of EMT-associated markers. These findings suggest that CD26 plays a functional role in promoting migratory activity of CRC cells and that its blockade can effectively attenuate EMT-driven cell migration. To further investigate the effect of the anti-CD26 antibody on the invasive potential of CRC cells, we performed a Transwell invasion assay using HCT-116 cells. The upper chambers of the inserts were coated with basement membrane matrix proteins to simulate the extracellular environment encountered during tumor invasion. Treatment with the anti-CD26 antibody significantly reduced the number of cells that invaded through the matrix-coated membrane compared with that in the control group (Figure 3D). These findings indicate that CD26 contributes not only to migratory activity but also to invasive behavior in CRC cells, likely through the promotion of EMT, and that its inhibition effectively suppresses both processes.

### 2.4. Anti-CD26 Antibody Treatment Suppresses EMT Signals

RNA was extracted after co-culturing HCT-116 cells with the anti-CD26 antibody, and the expression of epithelial markers was enhanced and that of mesenchymal markers was attenuated. E-cadherin expression increased, and N-cadherin expression decreased. To elucidate the molecular mechanism by which the anti-CD26 antibody regulates EMT, we evaluated the expression levels of important EMT markers at the mRNA and protein levels. Quantitative PCR revealed that treatment with the anti-CD26 antibody significantly increased the expression of the epithelial marker E-cadherin, and the mesenchymal-associated transcription factors ZEB1, Twist1, and Snail1 were markedly downregulated compared to the control group (Figure 4A). Consistent with the quantitative PCR results, Western blot analysis demonstrated an increase in E-cadherin protein expression and a concomitant decrease in N-cadherin levels following antibody treatment (Figure 4B,C). These findings indicate that the anti-CD26 antibody inhibits EMT by promoting epithelial characteristics and suppressing mesenchymal transition at the transcriptional and protein levels.

### 2.5. Anti-CD26 Antibody Suppresses Liver Metastasis

To evaluate the in vivo anti-metastatic effect of the anti-CD26 antibody, we used a liver metastasis model of BALB/c nude mice. HCT-116 CRC cells were injected into the spleens of mice, and liver metastasis was assessed 28 days after inoculation. Mice were treated with the anti-CD26 antibody or vehicle control according to the schedule shown in Figure 5A. Representative gross images of the liver at necropsy revealed a substantial reduction in visible metastatic nodules in the anti-CD26 antibody-treated group compared with that in the control group (Figure 5B). Quantitative analysis demonstrated that the number of metastatic lesions per liver and liver weight were significantly reduced in the antibody-treated groups (Figure 5C,D). These results indicate that anti-CD26 antibody administration effectively suppresses liver metastasis in vivo, likely through the inhibition of EMT and metastatic dissemination of CRC cells.

## 3. Discussion

This study demonstrates that targeting CD26 with a monoclonal antibody can suppress EMT and inhibit liver metastasis in CRC cells. CD26 (DPP-4) is a multifunctional transmembrane glycoprotein with known roles in immune regulation, enzymatic cleavage of peptides, and tumor progression [4,12]. In CRC, CD26 has been implicated as a marker of CSCs, particularly those with metastatic potential [2,3].

Our findings are consistent with those of the study by Pang et al., who first identified CD26^+^ cells in human CRC as a distinct subpopulation capable of initiating liver metastasis [3]. In support of its clinical relevance, analysis of publicly available datasets via GEPIA revealed that CD26 (DPP4) expression is significantly upregulated in tumor tissues compared to normal tissues in both colon and rectum adenocarcinomas. This finding is consistent with our in vitro observations and further implicates CD26 in colorectal tumor progression. In addition, they showed that CD26 expression correlated with poor patient survival, suggesting that CD26 is not merely a marker but a functional driver of metastasis. Moreover, Wesley et al. proposed that CD26 participates in tumor–stroma interactions and promotes invasive growth by modulating integrin and extracellular matrix dynamics [6].

In the present study, we observed that prolonged culture of HCT-116 cells led to an increase in CD26 expression along with EMT-related transcription factors ZEB1 and Twist1, which is consistent with previous findings that CD26 expression increases under conditions of high confluence or cellular stress [12,13]. Notably, treatment with an anti-CD26 monoclonal antibody reversed these changes, upregulating E-cadherin and downregulating ZEB1, Snail1, and Twist1 at the mRNA and protein levels. These factors are well-established transcriptional repressors of E-cadherin, acting through direct promoter binding or chromatin remodeling [14,15]. Their downregulation suggests a reversion of the EMT phenotype and restoration of epithelial characteristics.

Functionally, CD26 inhibition impaired the migratory and invasive capabilities of HCT-116 cells. The reduction in migration in the scratch wound-healing assay and suppression of invasion through basement membrane matrices in the Transwell assay reflect direct interference with EMT-mediated behaviors [16]. These observations are in agreement with the role of EMT as the central mechanism driving metastasis in solid tumors, including CRC [7].

Furthermore, we confirmed the anti-metastatic effect of CD26 inhibition in vivo using a well-established splenic injection model to mimic hematogenous dissemination to the liver [17]. Mice treated with the anti-CD26 antibody showed a significant reduction in hepatic metastases, supporting the idea that CD26 is functionally involved in metastatic colonization, possibly through its roles in extravasation, extracellular matrix remodeling, or evasion of immune surveillance [6,18].

Several mechanistic pathways may underlie the observed effects. CD26 regulates signaling pathways involved in EMT, including TGF-β, Wnt/β-catenin, and PI3K/AKT [19,20,21,22]. For example, CD26 interacts with caveolin-1, promoting downstream signaling such as NF-κB activation [23]. In addition, crosstalk among CD26 (DPP-4), caveolin-1, and integrin β1 has been implicated in TGF-β1-induced EMT signaling [23].

Clinically, these findings suggest that CD26-targeted therapies may provide a dual advantage by eradicating CSC populations and reversing EMT-associated phenotypes, thereby reducing metastatic recurrence. Although dipeptidyl peptidase inhibitors are widely used in the treatment of type 2 diabetes, their oncological application remains underexplored. However, recent efforts to repurpose or engineer CD26-specific antibodies, such as YS110, in mesothelioma and other malignancies show promising preclinical and early clinical activity [24,25,26,27,28]. Our study reinforces the rationale for developing CD26-targeted agents as part of combination strategies for the treatment of advanced CRC.

Limitations of this study include the use of a single CRC cell line and lack of mechanistic dissection of downstream signaling pathways. Further studies should examine whether CD26 inhibition alters TGF-β/Smad, β-catenin, or STAT3 signaling and whether these effects are dependent on CD26′s enzymatic activity or its scaffold function. Although the present findings were obtained using HCT-116 cells, previous studies have demonstrated that CD26 expression is regulated in a confluence-dependent manner in multiple CRC cell lines, including HCT-15 [11]. In addition, CD26 knockdown in HT-29 cells has been shown to suppress metastatic potential, while its overexpression in CD26-low cell lines such as DLD-1 and SW480 enhances metastatic behavior [29]. These findings support the broader relevance of CD26 in colorectal cancer. Nevertheless, validation in additional CRC cell lines will be important to confirm the generalizability of our observations. Furthermore, validation in patient-derived tumor models and immunocompetent systems would strengthen the translational potential of our findings. Although the mechanistic link between CD26 inhibition and metastatic suppression was not directly explored in this study, previous reports suggest that CD26 may promote metastasis through the CAV1/MMP1 signaling axis in colorectal cancer [29]. Given that EMT contributes to matrix remodeling and invasion, it is plausible that the anti-metastatic effect of the anti-CD26 antibody observed in our in vivo model may, at least in part, reflect interference with this pathway. Further studies will be required to determine whether the anti-CD26 antibody affects this axis directly or acts through broader suppression of EMT-related gene networks.

## 4. Materials and Methods

### 4.1. Cell Line and Cell Culture

The human CRC cell line HCT-116 was purchased from American Type Culture Collection (Manassas, VA, USA). HCT-116 cells were cultured in RPMI-1640 (Nacalai Tesque, Kyoto, Japan) supplemented with 10% fetal bovine serum (FBS; Biosera, Cholet, France) and antibiotics (100 U/mL penicillin and 100 μg/mL streptomycin) at 37 °C in a humidified atmosphere containing 5% CO_2_. All assays were performed in media containing 10% heat-inactivated FBS (56 °C for 30 min) to avoid serum-induced artifacts.

### 4.2. Antibodies

The mouse anti-human CD26 monoclonal antibody clone 1F7, which has been previously established by our group, was used in this study [30]. The monoclonal antibody 1F7 was developed via standard techniques after immunization of a BALB/c J mouse (The Jackson Laboratories, Bar Harbor, ME, USA) with cells of a PHA-stimulated T cell line derived from the New World primate species *Aotus triuirgatus*. Epitope mapping of 1F7 was performed in a previous study, revealing that IF7 binds to AA248-358th of the CD26 molecule [31].

### 4.3. Cytotoxic Assay

HCT-116 cells (5.0 × 10^3^ cells/well in a 96-well plate) were treated with the anti-CD26 antibody at concentrations of 1.17–75 μg/mL for 48 h, and cell viability was analyzed via the cell proliferation assay using alamarBlue™ cell viability reagent (Thermo Fisher Sientific, MA, USA). The medium was replaced with a reagent diluted to 10% in the medium and incubated for 3 h at 37 °C in a humidified atmosphere containing 5% CO_2_; afterwards, fluorescence was measured at 590 nm, with excitation at 560 nm, using a microplate reader (SpectraMax iD3, Molecular Devices, San Jose, CA, USA).

### 4.4. Scratch Wound-Healing Assay

HCT116 cells (2 × 10^5^ cells/well in a six-well plate) were cultured for three days until confluency, and a wound gap was created by gently scraping the cell monolayer with the tip of a P200 pipette (#WEF-200RS, BM Equipment Co., Ltd., Tokyo, Japan). Anti-CD26 antibody (15 μg/mL) was added and incubated at 37 °C in a humidified atmosphere containing 5% CO_2_. After 24 and 48 h, five wound gaps per well were photographed under a microscope (ECLIPSE Ts2, Nikon, Tokyo, Japan), and the relative distances were calculated.

### 4.5. Western Blotting Analysis

Cells were washed twice with phosphate-buffered saline (PBS) and then lysed with lysis buffer (50 mM HEPES, pH 7.5, 150 mM NaCl, 10% glycerine, 1 mM ethylene glycol tetraacetic acid, 1% Triton X-100, 1.5 mM MgCl_2_, and 1% protease inhibitor cocktail [Sigma Aldrich]). Total protein (40 µg) was fractionated via SDS-PAGE on a 4–20% polyacrylamide gel (Bio-Rad Laboratories, Hercules, CA, USA) using Tris-glycine SDS buffer (25 mM Tris, 192 mM glycine, 0.1% SDS). The separated proteins were subsequently transferred to polyvinylidene difluoride membranes (Thermo Fisher Scientific Inc., Waltham, MA, USA) for immunoblotting. Primary antibodies were used at the following dilutions and incubated overnight at 4 °C: anti-E-cadherin (3195T, 1:1000), anti-N-cadherin (13116T, 1:1000), and anti-β-actin (3700T, 1:1000). All primary antibodies were obtained from Cell Signaling Technology (Danvers, MA, USA). Secondary antibodies, such as anti-rabbit IgG horseradish peroxidase (HRP)-linked antibody (18-8816-31) from Rockland Immunochemicals, Inc. (Limerick, PA, USA), were used at 1:2000 ratio. β-actin was used for normalization. Immunoreactivity was visualized via chemiluminescence using an enhanced substrate for HRP detection (iBright CL1500, Thermo Fisher Scientific Inc.).

### 4.6. Gene Expression Analysis

Total RNA was isolated from pretreated cells using ReliaPrep™ RNA Miniprep Systems (Promega Co., Madison, WI, USA). RNA purity and concentration were determined based on the absorbance at 260 and 280 nm using a NanoDrop microvolume spectrophotometer (Thermo Fisher Scientific Inc.). First-strand cDNA synthesis from mRNA was performed using PrimeScript™ RT reagent kits (Takara Bio Inc., Kusatsu, Japan). Primer sequences are presented in the Appendix A. cDNA expression levels were determined using reverse transcription (RT)-quantitative polymerase chain reaction (PCR) (TP970 Thermal Cycler Dice® Real Time System III, Takara Bio Inc.). The expression level of each gene was quantified using the 2^−ΔΔCt^ method, with *GAPDH* as an internal control. All experiments were performed in triplicate.

### 4.7. Migration and Invasion Assays

A transendothelial migration assay using the CytoSelect™ Tumor Transendothelial Migration Assay kit (#CBA-216, Cell Biolabs. Inc., San Diego, CA, USA) was performed according to the manufacturer’s instructions. Briefly, HCT-116 cells (1 × 10^5^) were labeled with CytoTracker™ (fluorescence). Cells were suspended in 300 µL of serum-free RPMI-1640, stimulated with PBS or anti-CD26 antibody (15 µg/mL), and added to the upper inserts. RPMI-1640 (500 µL) with 10% FBS was added to the lower chamber. After 24 h, the cells that had migrated to the lower chamber were photographed at five locations in each well using a fluorescence microscope (BZ-X710, KEYENCE, Osaka, Japan), and the number of cells was counted and quantified.

The Transwell invasion assay was performed using the CytoSelect™ 24-well cell invasion assay kit (#CBA-111, Cell Biolabs. Inc.). Briefly, HCT-116 cells (1 × 10^5^) were suspended in 200 µL of serum-free RPMI-1640, stimulated with PBS or anti-CD26 antibody (15 μg/mL), and added to the upper inserts. RPMI-1640 (500 µL) with 10% FBS was added to the lower chamber. After 24 h, invaded cells in the lower chamber were used for fluorometric analysis, as per the manufacturer’s instructions. The fluorescence of each well was measured at an excitation wavelength of 480 nm and an emission wavelength of 520 nm using an automated microplate reader (SpectraMax iD3, Molecular Devices, CA, USA) with a compatible software program (SoftMax Touch (version 1.2.0.0), Molecular Devices).

### 4.8. In Vivo Tumor Metastasis in a Cancer Mouse Model

Female nude mice (BALB/cSlc-nu/nu, origin: Institute of Medical Science, University of Tokyo, total *n* = 24) were purchased from Japan SLC, Inc. (Hamamatsu, Japan). The mice were housed under a 12 h light/12 h dark cycle with free access to food and water and were allowed to acclimate for at least one week. The housing chamber temperature was maintained at approximately 23 °C. Tumor xenograft models were established via the injection of 5 × 10^5^ HCT-116 cells suspended in 0.1 mL Hank’s buffer in nude mice spleen. The protocol was approved by the Animal Care and Use Committee of the Juntendo University School of Medicine (approval number: 2023224). Euthanasia was performed under deep anesthesia using isoflurane inhalation, with the humane endpoint defined as the presence of intolerable suffering or a significant decrease in body weight (reduction of more than 20% compared to baseline). Anti-CD26 antibody or PBS as a control was intraperitoneally administered at 100 μg/body every three or four days starting from the day after transplantation. The mice were euthanized 28 days after transplantation, and their liver weights were measured and the number and size of metastatic lesions in the liver compared to examine the effect of anti-CD26 antibodies in suppressing colon cancer metastasis.

### 4.9. Statistical Analysis

Comparisons between the two groups were performed using Welch’s *t*-test. Statistical analyses were performed using Prism 10 version 10.0.0 (GraphPad Software, Boston, MA, USA), and differences were considered statistically significant at a *p*-value of <0.05.

## 5. Conclusions

We provide evidence that CD26 promotes EMT and liver metastasis in CRC stem-like cells and that targeting CD26 with a monoclonal antibody effectively suppresses these processes. These findings support CD26 as a promising therapeutic target for inhibiting EMT-driven metastatic progression in CRC.

## Figures and Tables

**Figure 1 ijms-26-07620-f001:**
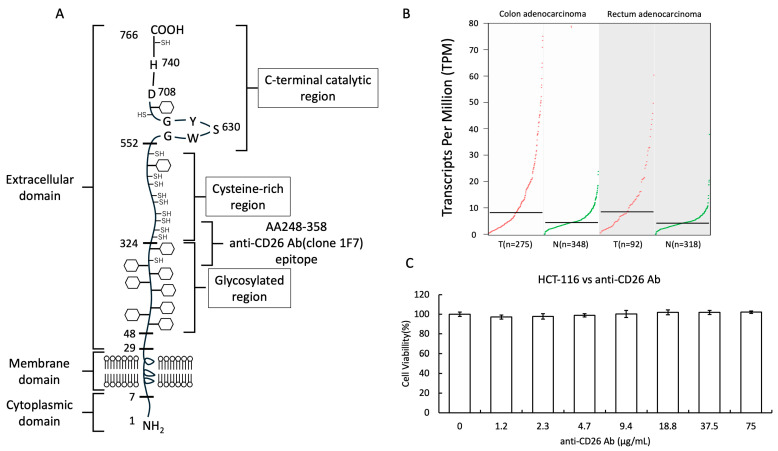
Structure of the CD26 molecule and cytotoxicity of anti-CD26 antibodies. (**A**) Schematic representation of the anti-CD26 antibody used in this study, highlighting the antigen-binding regions and domains responsible for the specific recognition of the extracellular domain of human CD26 (DPP-4). (**B**) Box plots showing CD26 mRNA expression in tumor versus normal tissues for colon adenocarcinoma and rectum adenocarcinoma, generated using the GEPIA (http://gepia.cancer-pku.cn/index.html, accessed on 25 July 2025) web tool based on TCGA and GTEx data. CD26 expression was significantly higher in tumor tissues compared to normal tissues. (**C**) Cell viability was assessed using the AlamarBlue assay following 48 h of treatment with increasing concentrations (0–75 μg/mL) of the anti-CD26 antibody. Data represent mean ± SD from three independent experiments. No significant cytotoxic effect was observed at concentrations up to 75 μg/mL, indicating negligible toxicity. DPP-4, dipeptidyl peptidase-4; T, tumor tissue; N, normal tissue; SD, standard deviation.

**Figure 2 ijms-26-07620-f002:**
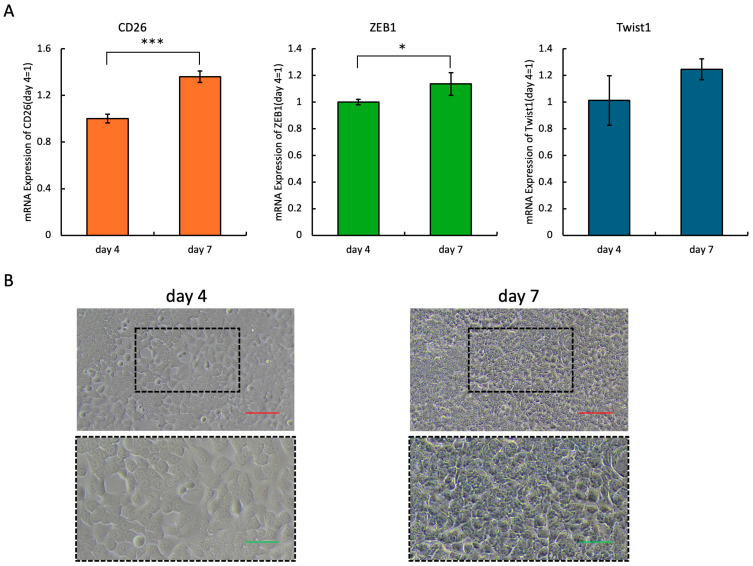
mRNA expression levels of CD26, ZEB1, and Twist1 in HCT-116 cells cultured for four and seven days. (**A**) Quantitative PCR analysis was performed to compare the relative mRNA expression of CD26, ZEB1, and Twist1 on days four and seven. Data are presented as mean ± SD from three independent experiments (* *p* < 0.05, *** *p* < 0.001 compared to day four). Expression of all genes significantly increased on day seven, indicating a potential relationship between prolonged culture conditions, enhanced CD26 expression, and EMT progression. (**B**) Phase-contrast microscopy images showing the confluence of HCT-116 cells on day four (left) and day seven (right). On day four, small intercellular gaps remained, whereas by day seven, the cells had reached full confluence with no visible spaces. The lower panels show magnified views of the dotted black rectangles in the upper images, highlighting the differences in cell density and morphology. Red scale bar = 100 μm, green scale bar = 50 μm. SD, standard deviation; PCR, polymerase chain reaction; EMT, epithelial-mesenchymal transition.

**Figure 3 ijms-26-07620-f003:**
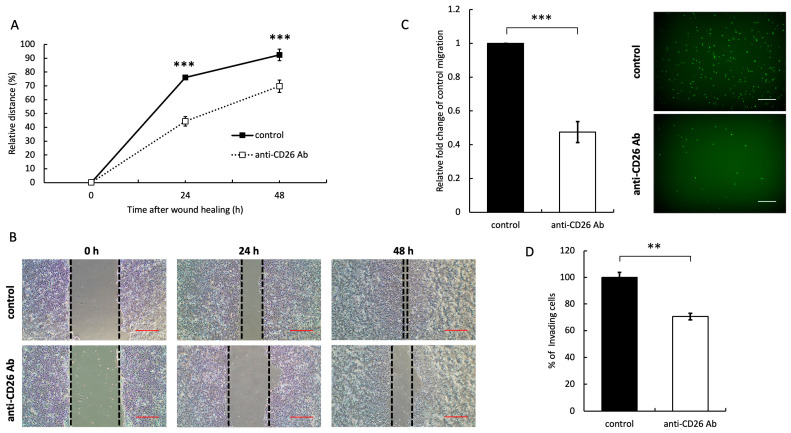
Anti-CD26 antibody treatment inhibits migration and invasion of HCT-116 cells. (**A**,**B**) Representative images and quantification of wound closure in HCT-116 cells treated with or without anti-CD26 antibody for 24 and 48 h. The antibody-treated group showed significantly reduced migration into the wound area compared to the control group. Data are presented as mean ± SD from three independent experiments (*** *p* < 0.001). Red scale bar = 500 μm. (**C**) HCT-116 cells were seeded in the upper chamber of a Transwell insert without Matrigel coating and treated with or without anti-CD26 antibody. After 24 h, migrated live cells on the lower surface of the membrane were fluorescently stained, and quantified. Representative images and quantification of migrated cells are shown. Data are presented as mean ± SD from three independent experiments (*** *p* < 0.001). White scale bar = 200 μm. (**D**) HCT-116 cells were seeded in Transwell chambers pre-coated with basement membrane matrix proteins and treated with or without anti-CD26 antibody for 24 h. After 24 h, live cells that had invaded the lower part of the chamber were fluorescently stained with CyQuant^®^ GR Dye, solubilized in lysis buffer, and the fluorescence intensity measured. The number of cells that invaded through the membrane was significantly reduced in the antibody-treated group compared with that in the control. Data are presented as mean ± SD from three independent experiments (** *p* < 0.01). These results suggest that CD26 inhibition suppresses cell motility, potentially through the attenuation of EMT. SD, standard deviation; EMT, epithelial-mesenchymal transition.

**Figure 4 ijms-26-07620-f004:**
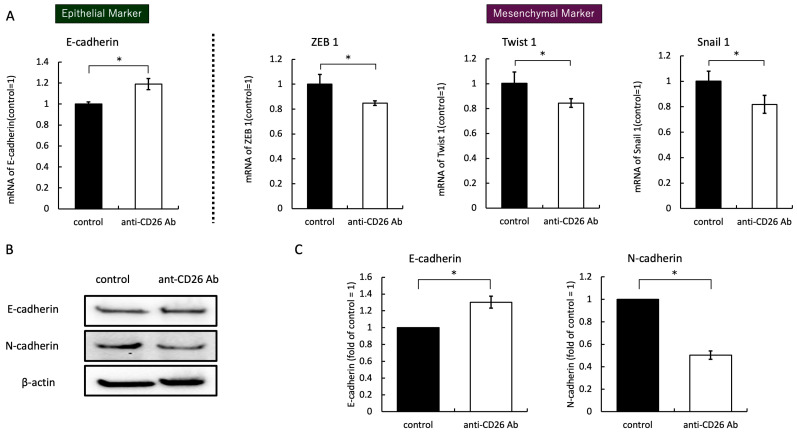
Anti-CD26 antibody modulates EMT-related gene and protein expression in HCT-116 cells. (**A**) Quantitative PCR analysis of EMT-related genes (E-cadherin, ZEB1, Twist1, and Snail1) in HCT-116 cells treated with or without anti-CD26 antibody for 24 h. E-cadherin expression was significantly upregulated, whereas that of ZEB1, Twist1, and Snail1 was downregulated following antibody treatment. (**B**,**C**) Western blot analysis of E-cadherin and N-cadherin protein expression under the same conditions. Anti-CD26 antibody increased E-cadherin levels and reduced N-cadherin expression, consistent with EMT suppression. Representative blots and quantification from three independent experiments are shown. Data are presented as mean ± SD (* *p* < 0.05). SD, standard deviation; PCR, polymerase chain reaction; EMT, epithelial-mesenchymal transition.

**Figure 5 ijms-26-07620-f005:**
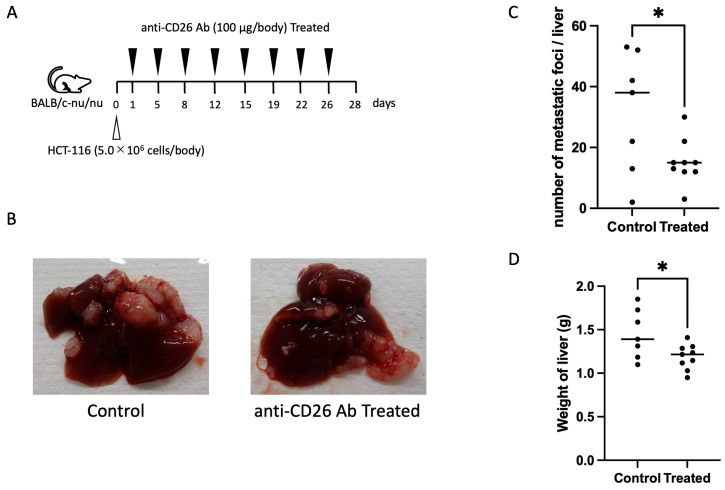
Anti-CD26 antibody suppresses liver metastasis of HCT-116 cells in a splenic injection mouse model. (**A**) Experimental timeline for the in vivo liver metastasis model. HCT-116 cells were injected into the spleens of BALB/c nude mice, followed by the administration of anti-CD26 antibody or vehicle control at indicated time points. Mice were sacrificed 28 days post-inoculation. (**B**) Representative gross images of livers from control and anti-CD26 antibody-treated mice at necropsy. The antibody-treated group showed visibly fewer metastatic nodules. (**C**,**D**) Quantification of metastatic lesions per liver and liver weight. The number of liver metastases was significantly reduced in the anti-CD26 antibody-treated group compared with that in the control group. Horizontal bars in the graph indicate the median values, and each point indicates the measured value for an individual (* *p* < 0.05).

## Data Availability

The original contributions presented in this study are included in the article. Further inquiries can be directed to the corresponding author.

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
