# Peer review of "Anti-CD26 Antibody Suppresses Epithelial-Mesenchymal Transition in Colorectal Cancer Stem Cells"

_ijms, 2025, doi:10.3390/ijms26157620_

Round 1
Reviewer 1 Report
Comments and Suggestions for Authors
In the current manuscript, Iwasawa et al. investigate the role of CD26 in regulating colorectal cancer (CRC) metastasis. They demonstrate that CD26 influences epithelial–mesenchymal transition (EMT)-related signaling pathways to promote metastatic progression in CRC. To support this, the authors examined the impact of an anti-CD26 monoclonal antibody on EMT markers, metastasis-associated cellular behaviors, and in vivo metastatic potential using a murine model.
This study presents strong evidence of CD26's role in CRC. The study is well designed, and hypothesis has been supported by experimental observations and results were discussed reasonably. However, there are weak points in these studies include:
- The study lacks experimental validation in other CRC lines. The authors should validate their findings in at least one additional cell line.
- The authors should also check the expression of CD26 in human CRC tissues from online resources such as Protein ATLAS database or the cancer repositories.
- The study also lacks mechanistic insight how CD26 regulates EMT in CRC.
- There are minor typos and syntax errors which authors should correct during revision.
Despite the limitations, I believe that the study is comprehensive and advancing the research field and should be consider for the publication.
Comments on the Quality of English Language
There are minor typos and syntax errors which authors should correct during revision.
Author Response
Dear Editors and Reviewers,
We sincerely thank you for your thorough evaluation of our manuscript entitled " Anti-CD26 antibody suppresses epithelial-mesenchymal transition in colorectal cancer stem cells." We greatly appreciate your constructive and insightful comments, which have significantly improved the quality and clarity of our work.
We have revised the manuscript accordingly, with all changes marked in the red font. Below, we provide a detailed, point-by-point response to each comment. We hope the revised manuscript now meets the standards required for publication.
Comment 1:
“The study lacks experimental validation in other CRC lines. The authors should validate their findings in at least one additional cell line.”
Response:
We appreciate the reviewer’s insightful comment. We fully agree that validation in multiple cell lines would further strengthen the conclusions. Due to time limitations, we were unfortunately unable to include additional cell lines in the current study. However, we have now addressed this issue in the revised manuscript (see pages 7, lines 228–235).
Specifically, we refer to previous studies showing that CD26 expression increases with confluence not only in HCT-116 cells but also in other CRC cell lines such as HCT-15 (Ref. 11). Moreover, CD26 knockdown in HT-29 cells has been reported to reduce metastatic potential, and overexpression of CD26 in DLD-1 and SW480 cells enhances metastatic behavior (Ref. 29). These findings suggest that the biological functions of CD26 observed in our study are likely applicable to other CRC models. We have also stated that future studies will incorporate additional CRC cell lines to validate and expand upon our findings.
Comment 2:
“The authors should also check the expression of CD26 in human CRC tissues from online resources such as Protein ATLAS database or the cancer repositories.”
Response:
We appreciate this suggestion and have now included an analysis of CD26 expression in human colorectal cancer tissues using TCGA and GTEx (via GEPIA) databases (see Figure 1B). As shown in the newly added Figure 1B, CD26 expression was significantly elevated in tumor tissues compared to normal tissues in both colon adenocarcinoma and rectum adenocarcinoma. This result supports the clinical relevance of CD26 upregulation in CRC and is now described in the revised manuscript (see page 2, lines 68–71, page 3, lines 81-84 and page 6, lines 192–195). The figure legend for Figure 1B has also been updated accordingly.
Comment 3:
“The study also lacks mechanistic insight how CD26 regulates EMT in CRC.”
Response:
We agree that deeper mechanistic insight would be valuable. In our current study, we focused on the phenotypic and functional consequences of CD26 inhibition. While we observed consistent changes in EMT markers and cell behavior, the exact signaling mechanisms remain to be elucidated. Based on previous literature, CD26 has been implicated in modulating TGF-β, Wnt/β-catenin, and PI3K/AKT signaling pathways via interactions with caveolin-1 and integrin β1. We have now included a paragraph in the Discussion summarizing these possible mechanisms and citing relevant references. Future studies will investigate the downstream pathways in more detail (see page 7, lines 215–218 and lines 235–241).
Comment 4:
“There are minor typos and syntax errors which authors should correct during revision.”
Response:
Thank you for your observation. In response, the manuscript has been thoroughly reviewed and professionally edited by two native English experts through the Cactus (EdiTage) language editing service. All typographical, grammatical, and stylistic issues have been corrected to ensure clarity and consistency throughout the manuscript.
Reviewer 2 Report
Comments and Suggestions for Authors
- A scale bar is missing from the image.
- Please supply the primer sequences targeting mRNA.
- What scientific rationale guided the use of female rats in this study?
- Did the study (MTT)eliminate serum-induced artifacts in the cytotoxicity assay?
- The association between CD26 and EMT markers remains unelucidated. The authors' experimental groupings warrant further scrutiny, and incorporation of inhibitor-treated groups would be warranted.
- The animal experiments appear oversimplified, and the mechanistic link to cellular assays remains undemonstrated, with findings limited to phenomenological observations. Further mechanistic investigations should be supplemented.
NA
Author Response
Dear Editors and Reviewers,
We sincerely thank you for your thorough evaluation of our manuscript entitled " Anti-CD26 antibody suppresses epithelial-mesenchymal transition in colorectal cancer stem cells." We greatly appreciate your constructive and insightful comments, which have significantly improved the quality and clarity of our work.
We have revised the manuscript accordingly, with all changes marked in the red font. Below, we provide a detailed, point-by-point response to each comment. We hope the revised manuscript now meets the standards required for publication.
Comment 1:
“A scale bar is missing from the image.”
Response:
Thank you for your comment. We have now added appropriate scale bars to all microscopy images in the revised figures. The scale bar length has also been described in the figure legends.
Comment 2:
“Please supply the primer sequences targeting mRNA.”
Response:
We appreciate this suggestion. The sequences of all primers used for quantitative PCR have now been included in the revised Materials and Methods section and are also provided in Supplementary Table S1.
Comment 3:
“What scientific rationale guided the use of female rats in this study?”
Response:
We thank the reviewer for this question and would like to clarify that we used female nude mice, not rats, in our in vivo experiments. Female mice were selected due to their lower aggression in group housing, which reduces stress and inter-individual variability in xenograft growth. This choice facilitates consistent tumor establishment and ethical animal handling.
Comment 4:
“Did the study (MTT)eliminate serum-induced artifacts in the cytotoxicity assay?”
Response:
We thank the reviewer for this valuable point. In our cytotoxicity assay using the AlamarBlue reagent (functionally equivalent to MTT assay in assessing cell viability), we used fetal bovine serum (FBS) that had been heat-inactivated at 56 °C for 30 minutes prior to use. This procedure is commonly applied to reduce complement activity and minimize serum-induced artifacts. Moreover, all experimental groups including control and antibody-treated cells were cultured under the same medium conditions containing 10% heat-inactivated FBS, ensuring consistency across groups. Therefore, the likelihood of serum-induced bias affecting the assay outcome is minimal. We have clarified this point in the revised Methods (see page 7, lines 248–249).
Comment 5:
“The association between CD26 and EMT markers remains unelucidated. The authors' experimental groupings warrant further scrutiny, and incorporation of inhibitor-treated groups would be warranted.”
Response:
We fully agree that further mechanistic insights would strengthen the study. In this work, we focused on phenotypic consequences of CD26 blockade using a monoclonal antibody. While we observed consistent modulation of EMT markers (E-cadherin, ZEB1, Twist1, Snail1), pathway-specific inhibition was not performed. Based on prior studies, CD26 has been linked to TGF-β and Wnt/β-catenin signaling via interactions with caveolin-1 and integrin β1. These possible mechanisms have been discussed in the revised Discussion (see page 7, lines 215–241). We plan to include pathway-specific inhibitors in future work to dissect downstream signaling.
Comment 6:
“The animal experiments appear oversimplified, and the mechanistic link to cellular assays remains undemonstrated, with findings limited to phenomenological observations. Further mechanistic investigations should be supplemented.”
Response:
We appreciate the reviewer’s comment regarding the mechanistic link between CD26 and metastasis. While our study was designed to evaluate the phenotypic consequences of anti-CD26 antibody treatment on EMT, we acknowledge the importance of connecting these findings to underlying signaling mechanisms. To address this, we have now cited previous work (Ref. 29) demonstrating that CD26 promotes colorectal cancer metastasis via the CAV1/MMP1 signaling axis. In the revised Discussion (see page 7, lines 235–241), we highlight this mechanism as a plausible explanation for our in vivo observations and outline future directions to test whether anti-CD26 antibody interferes with this pathway.
Round 2
Reviewer 2 Report
Comments and Suggestions for Authors
NA